# Preparation, Characterization and Application of Amorphized Cellulose—A Review

**DOI:** 10.3390/polym13244313

**Published:** 2021-12-09

**Authors:** Michael Ioelovich

**Affiliations:** Designer Energy, 2, Bergman St., Rehovot 7670504, Israel; ioelovichm@gmail.com; Tel.: +972-89366612

**Keywords:** amorphized cellulose, amorphous nanocellulose, preparation methods, structural organization, characterization, accessibility, reactivity, enzymatic digestibility, applications

## Abstract

This review describes the methods of cellulose amorphization, such as dry grinding, mercerization, treatment with liquid ammonia, swelling in solvents, regeneration from solutions, etc. In addition, the main characteristics and applications of amorphized celluloses are discussed. An optimal method for preparing completely amorphous cellulose (CAC) via the treatment of original cellulose material with a cold NaOH/Urea-solvent at the solvent to cellulose ratio R ≥ 5 is proposed. Structural studies show that amorphous cellulose contains mesomorphous clusters with a size of 1.85 nm and specific gravity of 1.49 g/cm^3^. Furthermore, each such cluster consists of about five glucopyranose layers with an average interlayer spacing of 0.45 nm. Amorphous cellulose is characterized by increased hydrophilicity, reactivity, and enzymatic digestibility. Due to its amorphous structure, the CAC can be used as a promising substrate for enzymatic hydrolysis to produce glucose, which can be applied in biotechnology for growing various microorganisms. In addition, the application of CAC in agriculture is described. A waste-free method for producing amorphous nanocellulose is considered, and the main applications of nanosized amorphous cellulose are discussed.

## 1. Introduction

Natural polysaccharides constitute a wide class of natural biopolymers that comprise cellulose, starch, dextran, agar, xanthan, xylanes, mannanes, galactans, alginates, carrageenans, etc. Most of these polysaccharides are low crystalline or amorphous and readily soluble in water [1]. A pronounced exception to this rule is natural cellulose, a highly crystalline and water-insoluble polysaccharide. As is known, cellulose is the most widespread natural biopolymer [2]. Natural cellulose is a structural constituent of all land plants and many algae [3]. In addition, cellulose occurs in shells (tunicates) of certain marine creatures [4], and this biopolymer is also produced by some microorganisms, e.g., *Gluconacetobacter xylinus* [5].

To isolate a pure biopolymer, the cellulose-containing raw material is treated with solutions of various chemicals, such as caustic soda, a mixture of sodium hydroxide with sodium sulfide, and sulfurous acid or its salts [6]. Some other pulping methods are also used on a small scale. The pulping process is followed by multi-stage bleaching.

Studies have shown that macromolecules of natural cellulose may contain 2000 to 30,000 repeating anhydroglucose units (AGU), which are linked by chemical β-1,4-glycosidic bonds [7,8]. In nature, cellulose chains are connected in the transverse direction by hydrogen and Van der Waals bonds and form nanofibrils, which contain crystallites and non-crystalline (amorphous) domains.

Crystallites of natural celluloses may exist in two different crystal forms, triclinic CIα and monoclinic CIβ. Crystallites of algae and bacterial cellulose have CIα allomorph, while crystallites in the cellulose of land plants and tunicates have CIβ allomorph [9,10]. After specific physicochemical treatments of natural cellulose, other crystalline allomorphs, CII, CIII, or CIV can be obtained [11].

Cellulose crystallites are stable constituents and therefore they are inaccessible to common organic solvents, water, and diluted water solutions of acids, alkalis, and salts [12]. The crystallinity, i.e., mass fraction of crystallites in natural cellulose, ranges from 0.53–0.55 (cellulose of herbaceous plants) to 0.77–0.80 (tunicate cellulose) [12,13]. Due to increased crystallinity, natural cellulose exhibits low accessibility to reagents. In particular, it gives a small yield of glucose after acidic [14] or enzymatic hydrolysis [15,16]. In addition, the high crystallinity of natural cellulose hampers dyeing, diffusion, and sorption of water vapor [17], which negatively affects its use in textile materials.

To increase the accessibility, reactivity, and sorption ability of natural cellulose, it is necessary to destroy the crystalline structure of this biopolymer to obtain amorphized or completely amorphous cellulose.

A literature search on this problem showed that there are only research articles devoted to preparation methods of amorphized cellulose by grinding [18] or processing with various solvents [19,20]. In addition, there are articles where some properties of amorphized cellulose are considered, e.g., acid hydrolyzability, thermal stability, etc. [21]. However, no reviews were found on amorphized cellulose. Therefore, it was necessary to collect and analyze the appropriate literature, including the author’s own results, and then summarize them in a special review.

The main purpose of this review article was to discuss the main methods of cellulose amorphization, to describe methods for determining the structural characteristics of initial and amorphized cellulose, as well as to study the effect of structural changes in cellulosic materials after amorphization on various properties, including sorption, hydrolyzability, and enzymatic digestibility.

## 2. Characterization Methods of Original and Amorphized Celluloses

Various methods can be used to study the structure and properties of cellulose materials and their structural changes during and after amorphization, as follows.

### 2.1. Wide-Angle X-ray Scattering (WAXS)

The WAXS method is designed to determine the position, intensity, shape, and width of the diffraction peaks of diverse materials, and to study features of crystalline structure, such as crystallinity, sizes of crystallites, type and content of crystalline allomorph, interplanar spacings, parameters of unit crystalline cells, lattice distortion degree, etc. X-ray diffractograms are recorded by a standard X-ray diffractometer using the CuK_α_-radiation (λ = 0.15418 nm). This diffractometer consists of X-ray tube, monochromator, slits to adjust the beam, goniometer, and detector. After recording the diffractograms, it is necessary to subtract the incoherent background and then correct the peak intensities. Corrections include the coefficient of X-ray absorption, LP coefficient, and Rietveld refinement. Corrected diffraction patterns are used to determine various structural characteristics of cellulose samples [22,23,24].

#### 2.1.1. Determination of the Angular Positions of the Main Diffraction Peaks

Parameters of crystalline structure of different allomorphs were shown in Table 1.

To calculate the interplanar spacings (d) in the crystalline lattice, Bragg’s equation is used:d = λ/(2sin θ_hkl_),(1)

#### 2.1.2. Determination of Crystallinity and Amorphicity Degree

The term “actual degree of crystallinity” means the mass fraction of the crystalline phase in the sample including fraction of paracrystalline layers of crystallites. To find the degree of crystallinity, mostly the X-ray method is used. This method requires the fulfillment of certain conditions: (1). The sample must be untextured; (2). The corrected X-ray diffractogram should be used, from which the scattering areas of crystalline (cr) and non-crystalline domains (nc) should be separated (Figure 1).

To calculate the actual degree of crystallinity (X), the area of crystalline scattering is divided by the full scattering area of the sample:(2)X=∫(Jc dφ)/∫(Jo dφ),
where J_c_ and J_o_ are the corrected and normalized intensities of X-ray diffraction of crystallites only and whole sample, respectively; φ = 2θ°.

The degree of amorphicity (Y) is calculated, as follows:Y = 1 − X,(3)

#### 2.1.3. Determination of Crystallite Sizes

To find an average size of small crystallites from X-ray patterns, Scherrer’s equation is usually used:D_sh_ = K λ/(B cos θ_hkl_),(4)
where B is the width of the peak at the half of the maximum of peak height (in radians); θ_hkl_ is Bragg’s angle at the peak maximum; shape coefficient K is usually taken close to 1.

However, Scherrer’s equation has limitations. First, this equation is not reliable if the crystallite size in any crystallographic direction reaches 100 nm, and secondly, it does not take into account the contribution of instrumental (b) and distortion (Δ) factors in the experimental width (B) of the peak [25,26,27]. Thus, to calculate the corrected width of the crystalline peak, the following equation should be used:B_o_ = (B^2^ − b^2^ − Δ^2^)^1/2^,(5)

The instrumental factor, b, can be measured using a crystalline standard, e.g., D-cellobiose. The contribution of the distortions in the width of the peak can be calculated as follows:Δ = 4δ tgθ_hkl_,(6)
where δ is distortion degree; d is interplanar spacings in the direction perpendicular to (hkl) planes of crystallites.

Thus, to calculate the actual size of crystallites, D_cr_, the updated equation should be used:D_cr_ = λ/(cos θ_hkl_) ((B^2^ − b^2^ − Δ^2^))^1/2^,(7)

For example, using the broadening of (200) peak of CIβ crystallites, the distortion factor, Δ, can be calculated by the following empirical equation [28]:Δ = 2.083d − 0.7997,(8)
where d (nm) is interplanar spacings in the direction perpendicular to 200-planes.

In this case, one can find the actual lateral size of the crystallites, D_cr_, in the direction perpendicular to the 200-planes.

The WAXS method can be also used to study the structural characteristics of completely amorphous cellulose. The X-ray diffraction pattern of such cellulose has a wide diffuse scattering with one maximum at 2θ_m_ ≈ 19.8°, which is typical for amorphous polymers having local ordering in the arrangement of repeating units. Based on the obtained diffraction pattern, the following structural characteristics of amorphous cellulose can be found. First, the average interplanar spacings (d_m_) in a distorted mesomorphous lattice can be calculated by Bragg’s Equation (1).
d_m_ = λ/(2sin θ_m_),(9)
where θ_m_ ≈ 9.9°.

Second, the average size (D_m_) of the mesomorphous cluster can be calculated using Warren’s equation for 2D structures [29].
D_m_ = k λ/(B_o_ cos θ_m_),(10)
where coefficient k = 1.84 and B_o_ is the corrected width of the diffuse peak.

Third, the electron and atomic density radial distribution functions can be calculated to study the near order in an amorphous matter [30].

### 2.2. Method of CP/MAS ^13^C NMR

The abbreviation CP/MAS ^13^C NMR denotes “Solid-state cross-polarization magic angle spinning ^13^C NMR spectroscopy”. The CP/MAS 13C-NMR spectra are recorded by spectrometers operating at 50–75 MHz ^13^C resonance frequency. A zirconium oxide rotor is used. The MAS rate is 4–5 kHz. The acquisition is made with a CP pulse sequence using a 3–4 μs proton 90° pulses; 0.8–1 ms contact pulse and a 2.5–3 s delay between repetitions. Accumulation numbers of the pulses are >2000 to provide the total acquisition time ≥2 h. Glycine can be used as an external standard for the calibration of the chemical shift scale relative to tetramethylsilane. The data point of maximum intensity in the glycine carbonyl line is assigned a chemical shift of 176.03 ppm.

The CP/MAS ^13^C NMR method generates peaks (signals) from carbon atoms of anhydroglucose unit of cellulose [31,32]. Atoms C1 give a peak in the range of chemical shift (δ) of 100–110 ppm, atoms C4-in the range of 82–90 ppm, C2, C3, and C5 in the range of 70–80 ppm, and atoms C6 in the range of 60–68 ppm (Figure 2).

To estimate the index of crystallinity (CrI), the “crystalline” and “non-crystalline” peaks of C4 atoms are usually used. The peak at 88–89 ppm is attributed to C4 atoms of crystallites, while the peak at 83–84 ppm to C4 atoms of non-crystalline domains. Indeed, after amorphization of the crystalline cellulose, the “crystalline” peak at 88–89 ppm disappears. A problem is that the C4 peaks at 83–84 ppm and 88–89 ppm are overlapping. To separate these “crystalline” and “non-crystalline” peaks, a special deconvolution procedure was performed. Then, the CrI was calculated by the equation:CrI = F_cr_/(F_cr_ + F_nc_),(11)
where F_cr_ and F_nc_ are areas of peaks (signals) attributed to C4-atoms of crystallites and non-crystalline domains, respectively.

### 2.3. Sorption of Water Vapor

Sorption isotherms of water vapor by cellulosic materials have a sigmoidal shape, similar to multilayer adsorption isotherms. However, the BET equation cannot be used for the calculation of sorption characteristics of cellulose, because the sorption mechanism of water vapor in this biopolymer is no adsorption of the sorbate on the surface of pores, but absorption of the vapor molecules into non-crystalline domains (NCD) of cellulose. In this case, a thermodynamic Equation (12) is used to describe the sorption isotherms [33]:S = S_o_ Y/(1 − K lnα),(12)
where S is the sorption value at a temperature of 25 °C and certain relative vapor pressure, α = p/p_o_; S_o_ = 0.507 is the maximum sorption value for completely amorphous cellulose at α = 1; K = 2.7 is coefficient; Y is the degree of amorphicity.

Thus, at constant α-value, the sorption of water vapor is directly proportional to a fraction (Y) of NCD in the cellulose sample.

### 2.4. Heat of Wetting

Calorimetry is a thermodynamic method that provides the determination heat effect of the interaction of cellulose samples with various liquids. For this study, various types of calorimeters and microcalorimeters can be used [34]. The standard heat effect of interaction is measured at 25 °C and pressure of 1 atm. The experiments show that the interaction of dry cellulose samples with water at standard conditions is accompanied by the exothermic heat of wetting (Q). Because the interaction with water is carried out in non-crystalline domains (NCD) of cellulose, the heat of wetting is directly proportional to a fraction (Y) of NCD in the cellulose sample:Q = Q_o_ Y,(13)
where Q_o_ = 168 J/g is the maximum heat of wetting for completely amorphous cellulose in a dry state and Y is the degree of amorphicity.

## 3. Methods of Cellulose Amorphization

Various physical and physicochemical methods can be used to destroy the crystalline structure of cellulose. Among physical methods, dry grinding of cellulose in ball mills is most widespread [18,35,36]. For this purpose, cellulose sheets are dried, cut into small pieces, and subjected to preliminary disintegration in a knife mill. The resulting dry fibers are ground for 24 h, at least, in a ball mill using porcelain balls. The main problem of using ball milling for the amorphization of cellulose is the high cost of this process, estimated at $10–15 per kg. In addition, the obtained amorphized cellulose has an unstable phase state and recrystallizes in a humid atmosphere or aqueous medium [36].

Mercerization is the treatment of cellulose with a concentrated (18–20%) solution of sodium hydroxide. This treatment is accompanied by partial decrystallization, as well as a transformation of crystalline allomorph CI into allomorph CII [37]. Due to these structural changes, mercerization is widely used in physicochemistry and chemistry of cellulose to improve the dyeing, for determining the content of alpha-fraction, and for the preliminary activation of cellulose structure before etherification.

The treatment of cellulose with liquid ammonia, primary amines, and ethylenediamine (EDA) causes both partial decrystallization and transformation of crystalline allomorph CI into allomorph CIII [7,38,39]. However, neither treatment with alkali, nor treatment with liquid ammonia, amines, or EDA leads to complete amorphization of cellulose.

It is also known that after the regeneration of cellulose from solutions, its structure is amorphized. For this purpose, you can use various cellulose solvents, such as ionic liquids (IL), NMMO, LiCl/DMAA, DMSO/PFA, DMSO/DEA/SO_2_, H_3_PO_4_, etc. [40,41,42,43,44,45]. However, the application of these solvents for the amorphization of cellulose has a significant shortcoming, the quite high cost of such solvents. So, according to the Alibaba catalog, the average price of DMSO, DEA, DMAA, and H_3_PO_4_ is $1.5–2 per kg, NMMO $10–20 per kg, and IL $100–150 per kg. In addition, the mentioned organic solvents are toxic substances, while H_3_PO_4_ is an inorganic acid irritating the skin and eyes. The traditional cellulose solvents, CS_2_/NaOH system, and Cuproxam, used in the 20th century for the production of artificial cellulose fibers and films, are currently prohibited due to their toxicity and environmental hazard.

The cheapest cellulose solvent is probably an aqueous solution of 7% NaOH/12% Urea [46,47] denoted as N/U, which has a cost of $0.05 per kg due to the low price of commercial chemicals. Urea is a harmless substance and sodium hydroxide is a non-toxic but irritating substance that nevertheless is widely used in compliance with safety rules. The problem is that only the dilute cellulose solution in this solvent can be obtained, which complicates the regeneration process and reduces the productivity of amorphized cellulose (AC). To overcome this shortcoming, the use of the cellulose swelling process in the N/U solvent instead of dissolution was proposed [48].

The initial cellulosic pulp was placed in a glass beaker cooled with an ice/salt mixture to a temperature of −15 °C. Then, a cold N/U solvent was added at the ratio of the solvent to cellulose material (R) from 3 to 10 (*v*/*w*) while periodically stirred for 1 h, and after which left overnight in a freezer at −15 °C. The treated samples were removed from the freezer and mixed with a 10-fold volume of tap water. The swollen, gel-like samples were separated from a liquid medium on a vacuum glass filter, washed with water, neutralized with 1% HCl to pH 6–7, and then again washed with water. To study the structural characteristics and some physicochemical properties, the wet samples were rinsed with ethanol and dried at 60 °C to constant weight.

Along with the direct methods of obtaining amorphized cellulose, there are also indirect methods, e.g., by saponification of cellulose acetate in non-aqueous media [49]. The resulting amorphous cellulose has an unstable phase state and crystallizes in a humid atmosphere or aqueous medium.

A special type of amorphous cellulose is amorphous nanocellulose (ANC) dispersed to nanosized particles. It is usually obtained through acid hydrolysis of regenerated cellulose with subsequent ultrasound or mechanical disintegration in an aqueous medium [50]. This method comprises stages of cellulose dissolution, regeneration from solution to obtain amorphized cellulose (AC), partial acid hydrolysis of AC, washing of depolymerized AC, dilution with water, and comminuting in an aqueous media using ultrasound disintegrator, high-pressure homogenizer, microfluidizer, and some other apparatus. The main shortcomings of the common ANC production method are the loss of the solvent and part of cellulose, high consumption of chemicals and energy, as well as low productivity.

To overcome these shortcomings, a novel waste-free technology must be used [51]. This technology includes dissolving cellulose in sulfuric acid at low temperatures, holding for partial hydrolysis and reducing the degree of polymerization, regeneration of AC by dilution with water, separation of AC from the acid solution, washing and disintegration of AC in aqueous media to obtain nanoparticles, use of dilute acid for the production of by-products, and return of used water to the production cycle.

The general technology can be detailed by the following specific example. The initial pulp was mixed in a beaker with the predetermined amount of water. Then, 80 wt. % sulfuric acid was slowly added while cooling in an ice-water bath and stirring to obtain the required final acid concentration of 66% and acid/cellulose ratio of 10. The beaker was placed into the ice-water bath and stirred for 60 min. After acidic treatment, the contents of the beaker were poured out into a threefold volume of cold water while stirring to regenerate flocs of amorphous cellulose (AC). The flocs were separated from the dilute acid by centrifugation at 5000× *g* for 10–15 min; washed with distilled water to a pH of about 5, separating them from water by centrifugation. The separated flocs of AC were diluted with distilled water to a concentration of 1% and disintegrated by an ultrasound disperser “Branson S450CE” at 20 kHz for 10–15 min. The 1 wt. % water dispersion of ANC was evaporated under vacuum at 50 °C in order to obtain a hydrogel with a solids content of 10 wt. %.

The dilute acid and acidic washing water were collected together and neutralized with cheap Ca-containing compounds, e.g., calcium hydroxide, to precipitate calcium sulfate (CaSO_4_) [51]:H_2_SO_4_ + Ca(OH)_2_ → CaSO_4_ + 2H_2_O

This by-product is widely used as a white pigment for paints, filler for polymers and paper compositions, as well as an inorganic binder for the production of putties, plasters, and drywall. Another cheap Ca-containing compound can be mineral-hydroxylapatite, Ca_5_(PO_4_)_3_OH. As a result of treatment of the dilute acid and acidic washing water with hydroxylapatite, the sulfuric acid is almost completely utilized and turns into such valuable by-product as superphosphate fertilizer:7H_2_SO_4_ + 2Ca_5_(PO_4_)_3_OH → 3Ca(H_2_PO_4_)_2_ × 7CaSO_4_ + 2H_2_O

The wastewater was purified and returned to the technological cycle. Thus, the proposed waste-free technology ensures the production of ANC and complete utilization of solvent and reagent-sulfuric acid, by its conversion into valuable by-products, the sale of which can significantly reduce the cost of ANC. Studies have shown that ANC can be obtained with a yield of 60–65%. Particles of ANC have a round shape and average diameter of about 100 nm [50] (Figure 3).

The isolated ANC is characterized by a high amorphicity degree and decreased degree of polymerization, DP (Table 2). Moreover, the ANC contains sulfonic and reducing aldehyde groups.

## 4. Characterization of Amorphized and Amorphous Cellulose

### 4.1. Structural and Physico-Chemical Characteristics of Amorphized Cellulose

Some structural characteristics of AC samples are shown in Table 3.

The obtained results showed that after dry milling, mercerization, or treatment with liquid ammonia, the complete amorphization of cellulose does not occur.

After cellulose processing with a cold N/U solvent at R = 3, the CI allomorph of the initial cellulose is transformed into a CII allomorph with a low crystallinity degree. However, when the initial cellulose is treated with the cold solvent at R ≥ 5, completely amorphous cellulose (CAC) is formed (Figure 4).

Thus, the minimum R-value required to prepare the CAC is about 5. The same minimum R-value is required to obtain CAC by the treatment of mixed waste paper (MWP) containing waste paper towels, paper wipes, and blotting/absorbent paper. The structural changes of cellulose after various treatments were also analyzed using physicochemical methods (Table 4).

As follows from the obtained results (Table 4, Figure 5 and Figure 6), the most amorphized samples (Y = 1) are also the most hydrophilic.

### 4.2. Structural Features of Completely Amorphous Cellulose

The X-ray scattering of completely amorphous cellulose (CAC) has a wide reflex with a maximum at 2θ ≈ 19.8° (Figure 4), which is typical for amorphous polymers having local ordering in the arrangement of repeating units [52]. Such structural organization can be called “mesomorphous” because its order is intermediate between highly ordered crystalline and completely disordered amorphous structures [53].

Structural studies showed that amorphous cellulose contains mesomorphous clusters [54]. The average spacing between planes in such clusters, d_m_ ≈ 0.45 nm, was calculated by Bragg’s Equation (1), taking into account that θ ≈ 9.9°. In addition, using the Warren equation for 2D structures, the average size of the mesomorphous cluster was calculated, D_m_ ≈ 1.85 nm. These nano-sized mesomorphous clusters can serve as local dots for the subsequent growth of crystallite nuclei.

Further, the function of radial distribution of electron density (F) was also calculated to refine the cluster structure of CAC [55]:F = 2r/π ∫ si(s) sin (rs) ds(14)
where i(s) is corrected reduced intensity and s = (4π/λ) sin θ.

With increasing radius r, the cyclic F-function gradually fades. Nevertheless, the graph of this function until completely fading has five maximums of electron density at r-values of 0.1, 0.45, 0.9, 1.4, and 1.9 nm (Figure 7).

The first maximum of F-function at 0.1 nm is intramolecular and relates to the superposition of bond lengths between atoms in repeating anhydroglucose units (AGU) of amorphous cellulose. The second maximum of electron density is intermolecular and corresponds to average spacing d_m_ = 0.45 nm between planes in the mesomorphous cluster, whereas the third and fourth maximums correspond to the doubled and tripled d_m_ -value. The radius of electron density at 1.9 nm correlates with the average size of the mesomorphous cluster, D_m_. As is follows from calculations, the cluster of CAC can consist of about five layers formed by AGU.

Based on the results of structural studies, a model of the mesomorphous unit cell of the cluster in AC was proposed (Figure 8).

Such a model has angle γ of about 120° and parameters a and b of about 0.9 nm. Average interlayer distance, d_m,_ can be calculated, as follows:d_m_ = b sin[(180 − γ)/2] = 0.45 (nm),(15)

In this model of the mesomorphous cluster of CAC, the layers formed by AGU are oriented along 110-directions. Only one distance (d_m_) between 110 -planes in all cells is constant: d_m_ = 0.45 nm. Consequently, only one maximum can be present on the X-ray pattern of amorphous cellulose, at 2θ ≈ 19.8°, corresponding to interlayer spacing, d_m_.

Interplanar distances in other directions for different cells are not constant and can vary in limited range owing to the irregular shifting of layers, which hinder the appearance of other diffraction maximums for amorphous cellulose. Furthermore, the specific gravity of the mesomorphous clusters (ρ_m_) was evaluated from the known equation for rhombic unit cell:ρ_m_ = C/(a b c sin γ),(16)
where C = 1.076 is the coefficient of dimension, a = b = 0.9 nm, c = 1.03 nm, and γ = 120°.

The calculated specific gravity of the mesomorphous clusters, ρ_m_ ≈ 1.49 g/cm^3^, is slightly higher than the average specific gravity of amorphous cellulose, ρ_a_ ≈ 1.44–1.45 g/cm^3^ [12] (see also Appendix A).

## 5. Acidic Hydrolyzability and Enzymatic Digestibility

Cellulose samples can be hydrolyzed by acids and cellulolytic enzymes. To evaluate the hydrolyzability of cellulose, the cellulose samples were treated with 2.5 M sulfuric acid at 100 °C for 4 h using acid to sample ratio of 40 [54]. The hydrolyzate was separated from residue by centrifugation, and the acid was neutralized with calcium carbonate. The formed precipitate of calcium sulfate was separated from the aqueous phase by centrifugation, after which the liquid was analyzed to determine the concentration of glucose.

To study enzymatic digestibility, the cellulose samples were treated with a commercial cellulolytic enzyme Cellic CTec-3 (Novozymes A/S, Bagsvaerd, Denmark) [48]. The dose of the enzyme was 30 mg per 1 g of solid sample. Hydrolysis of the samples was carried out in 50-mL polypropylene tubes. The samples containing 1 g of solid matter and 1 mL of 50 mM acetate buffer (pH = 4.8) were put into the tubes. Then, the needed amount of the enzyme was added. Further, an additional amount of the buffer was supplemented to obtain a concentration of the cellulose substrate from 50 to 150 g/L. The tubes closed with covers were placed in a shaker incubator at 50 °C and shaken for 48 h. The hydrolyzate was separated from residue by centrifugation, after which the liquid was analyzed to determine the concentration of glucose.

The concentration of glucose in hydrolyzate after acidic or enzymatic hydrolysis was determined by the HPLC-apparatus of Agilent Technologies 1200 Infinity Series. The Amines HPX-87H column was used. The main conditions of the analysis were temperature 45 °C; mobile phase 0.005 M sulfuric acid; flow rate 0.6 mL/min. The hydrolyzate was preliminarily filtered through a 0.45 μm Nylon filter. The yield of glucose (YG, %) after hydrolysis was calculated by the equation:YG = 100% (C_g_/C_o_),(17)
where C_g_ is the final concentration of glucose in hydrolyzate (g/L) and C_o_ is the initial cellulose content (g/L).

The following samples were used to test the acidic hydrolyzability and enzymatic digestibility:Original chemical-grade cotton cellulose (ORC) having 98% α-fraction and DP = 2700Mercerized cellulose (MRC)Ammonia treated cellulose (AMC)Ball-milled cellulose (BMC)Cellulose materials treated with N/U-solvent at R = 5 and 10, which are completely amorphous cellulose samples (CAC)

The study of acidic hydrolyzability showed (Figure 9) that after the amorphization of original cellulose by different methods, a significant increase in the glucose yield is observed, especially for completely amorphous cellulose samples (CAC). This phenomenon can be explained by a directly proportional relationship between the hydrolyzability and the amorphicity degree of cellulose (Figure 10).

An exception is observed for two samples, AMC and BMC, which have an unstable phase state and partly crystallize under relatively mild reaction conditions (processing in an aqueous medium at 100 °C), and thereby their acidic hydrolyzability is reduced. In contrast to these two samples, the rest of the studied samples, including CAC and other amorphous samples obtained by regeneration from cellulose solutions, do not crystallize under the said mild reaction conditions [49]. To crystallize CAC samples via processing in the aqueous medium, a higher temperature is required.

Since the method of cellulose swelling in N/U solvent proved to be the most effective to increase its acidic hydrolyzability, this method was also applied for pretreatment of cellulose samples intended for enzymatic hydrolysis. For such pretreatment, samples of original cotton cellulose (ORC), microcrystalline cellulose Avicel PH-101 (MCC), and mixed waste paper (MWP) were used as initial materials.

Without solvent pretreatment, the initial materials have a moderate enzymatic digestibility at C_o_ = 50 g/L with YG 44–48%. However, after swelling in a cold N/U-solvent at the solvent to cellulose ratio R ≥ 5, the obtained samples are hydrolyzed almost completely (Figure 11) due to the amorphous structure of these samples (Table 3).

Despite the high enzymatic digestibility at the content of CAC, C_o_ = 50 g/L, the glucose concentration in hydrolyzate does not exceed 55 g/L (5.5%), which is too low for further fermentation. To increase the concentration of glucose, the initial content of the CAC substrate should be enhanced. If C_o_ is increased to 150 g/L, then glucose concentration in the hydrolyzate increases to 147 g/L (Figure 12).

## 6. Potential Applications of Amorphized Cellulose

Currently, amorphized cellulose (AC) is used only in laboratory research. For example, AC obtained by ball-grinding is used as an amorphous standard for structural studies of cellulose [16]. Another example is AC regenerated from a solution in o-phosphoric acid, which is used as an amorphous standard in the enzymatic hydrolysis of cellulose and plant biomass [45].

There are two main reasons that hinder the commercialization of amorphized cellulose. The first is the high cost of the original cellulose. So, the market price of bleached pulp is $600–800 per ton, whereas the price of cotton cellulose is even higher and reaches $1000 per ton. Therefore, it is not profitable to use them to produce cheap amorphous cellulose on a pilot or industrial scale.

The second reason is the high consumption of chemicals and energy for most of the methods of AC production. The only exception is the amortization method by swelling of the starting material in a cold N/U solvent [48]. This process is the cheapest of all known due to the low cost of chemicals and low energy consumption.

For commercial production of amorphous cellulose, it is most advisable to use cheap raw materials such as mixed waste paper (MWP), which can be supplied for $50 per ton. Studies have shown that if the content of MWP-based AC substrate in the reaction system reaches C_o_ = 150 g/L, then after 48 h of hydrolysis at the enzyme dose of 3%, a quite high glucose concentration C_g_ = 141 g/L can be achieved [48]. As a result, quite cheap glucose can be released. This technical-grade product can find diverse applications in biotechnology to produce ethanol [56]; acetic, lactic, and citric acids [57]; proteins, yeasts, and enzymes [58,59,60]; bacterial cellulose [61], polyhydroxyalkanoates [62], and other valuable bioproducts.

If MWP, after swelling in a cold N/U solvent, is washed with water, but not dried, then the resulting technical AC takes the form of a hydrogel. As is known, hydrogels are capable of holding large amounts of water in their three-dimensional networks [63,64]. The hydrogels can be used in biology, chemistry, physical chemistry, chromatography, medicine, hygiene, pharmaceutics, food processing, and other areas [65,66].

Recently, much attention has been paid to the use of hydrogels in agriculture and soil technology [67,68,69,70,71,72]. However, agricultural applications are limited by the high cost of synthetic polymeric hydrogels, their low production volume, and biostability.

To solve these problems, it is necessary to use cheap, available, and biodegradable natural material, such as MWP, for the production of hydrogels. Such hydrogels can be produced by the swelling of MWP in a cold N/U-solvent at R ≥ 5 followed by washing [73]. Another technology of hydrogel aimed especially for growing plants was the following: after MWP swelling in the cold N/U solvent, the swollen sample was treated with phosphoric acid to obtain a hydrogel containing phosphate salt and urea serving as PN fertilizer. This hydrogel enriched with PN fertilizer is highly effective in seed germination, and after usage, it can be completely decomposed under the action of microorganisms present in the soil [73].

Undried amorphous nanocellulose (ANC) can also form a hydrogel. It was found that the hydrogel of ANC can be used as an effective thickener. Therefore, a small addition of this hydrogel significantly increases the phase stability of aqueous dispersions of some drugs, e.g., “Maalox” (Figure 13).

Particles of ANC can also find application in medicine and cosmetics as a carrier of therapeutically active substances (TAS). Due to the increased content of acidic sulfonic and reducing functional groups (Table 2), ANC can serve as a nano-carrier of various TAS. For example, sulfonic groups of ANC can attach some bactericides, e.g., Ag-cations and ZnO [74]:ANC-SO_3_H + Ag^+^ → ANC-SO_3_Ag + H^+^
ANC-SO_3_H + ZnO → ANC-SO_3_ZnOH

In addition, acidic functional groups of ANC contribute to the ion binding of amino acids containing basic amino groups, e.g., arginine, lysine, or histidine, by the following scheme:ANC-SO_3_H + H_2_N-A → ANC-SO_3_^− +^H_3_N-A

In particular, the obtained ANC-Arginine conjugate can be used for the effective treatment and regeneration of damaged skin tissues, cure of herpes simplex virus, and in other arginine-therapy areas [75,76].

The presence in ANC reducing groups provides additional opportunities for attaching TAS. For example, the joining of a proteolytic enzyme like trypsin to nanoparticles of ANC can be expressed by the scheme:ANC-HC=O + H_2_N-R → ANC-CHOH-NH-R

Along with trypsin, some others enzymes, such as chymotrypsin, papain, collagenases, lysoamidases, lysozymes, etc., can be attached to ANC–nanocarrier and used for the treatment of wounds and burns [77].

## 7. Conclusions

Cellulose is the most abundant and widely used crystalline biopolymer. However, due to increased crystallinity, cellulose exhibits low accessibility to reagents. In particular, it gives a small yield of glucose after acidic or enzymatic hydrolysis. To increase the accessibility and reactivity of natural cellulose, it is necessary to destroy the crystalline structure of this biopolymer. Various physical and physicochemical methods can be used for the amorphization of cellulose structure, such as dry grinding, mercerization, treatment with liquid ammonia, primary amines, EDA, and diverse solvents, etc. The problem is that most of these methods are quite expensive and cannot be used for cellulose decrystallization on a large scale.

The exception is a quite cheap solvent that is an aqueous system of 7% NaOH/12% Urea (N/U). It has been found that after cellulose treatment with cold N/U-solvent at the solvent to cellulose ratio R ≥ 5, a completely amorphous sample (CAC) can be obtained. Structural studies showed that amorphous cellulose contains mesomorphous clusters with an average size of 1.85 nm and specific gravity of 1.49 g/cm^3^. Furthermore, each cluster consists of about five glucopyranose layers with an average interlayer spacing of 0.45 nm.

Amorphous cellulose is characterized by increased hydrophilicity, accessibility, reactivity, and enzymatic digestibility. Due to its amorphous structure, the CAC is converted to glucose almost completely in a short time under the action of a relatively small dose of cellulolytic enzymes. Such a CAC sample can be used as an amorphous standard, and as a substrate for the commercial production of glucose, which can find application in biotechnology as a promising nutrient for various microorganisms. In addition, the hydrogel of amorphous cellulose can be applied in agriculture.

The amorphous nanocellulose (ANC) having round particles with an average diameter of about 100 nm can be prepared using waste-free technology. The hydrogel of ANC can be used as a thickener. Moreover, due to the presence of acidic sulfonic and reducing functional groups, ANC can serve as a nano-carrier of various therapeutically active substances.

## Figures and Tables

**Figure 1 polymers-13-04313-f001:**
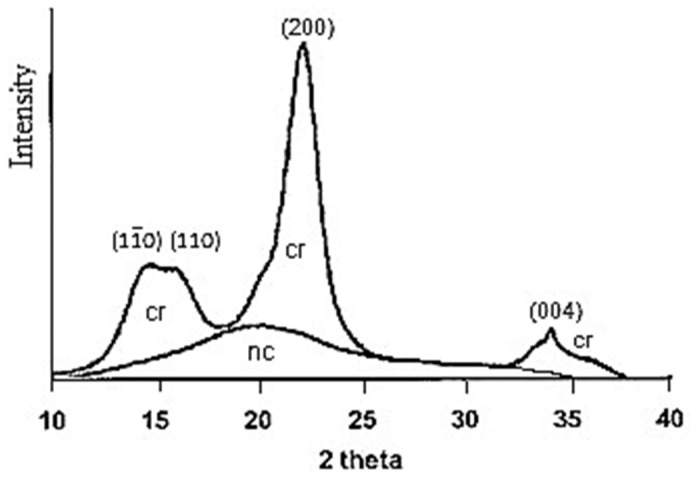
Illustration of method for separation of scattering regions related to crystalline (cr) and non-crystalline (nc) domains.

**Figure 2 polymers-13-04313-f002:**
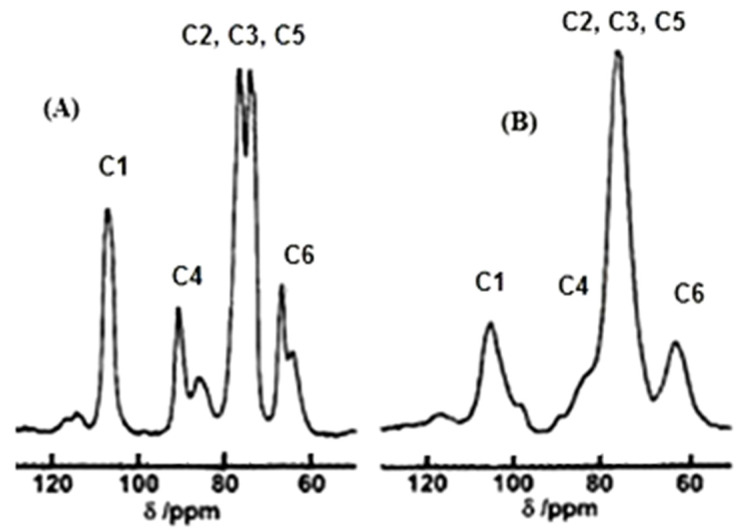
Solid-state ^13^C NMR spectrum of crystalline (**A**) and amorphous (**B**) cellulose.

**Figure 3 polymers-13-04313-f003:**
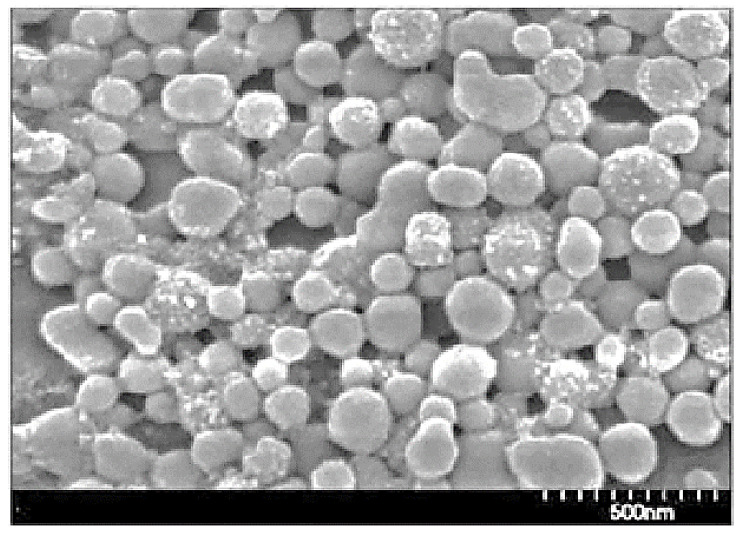
SEM image of nanoparticles of ANC.

**Figure 4 polymers-13-04313-f004:**
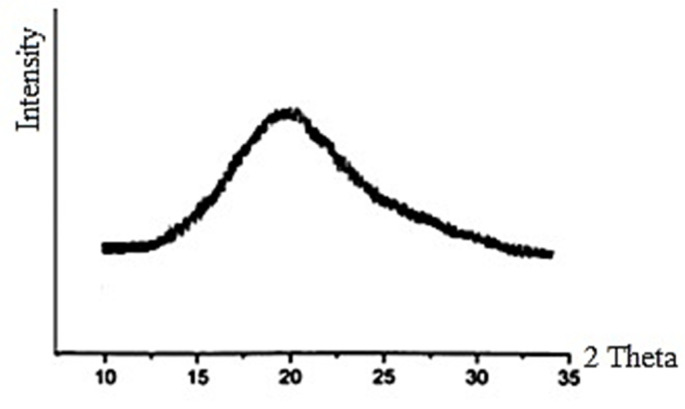
Illustration of X-ray diffractogram of completely amorphous cellulose.

**Figure 5 polymers-13-04313-f005:**
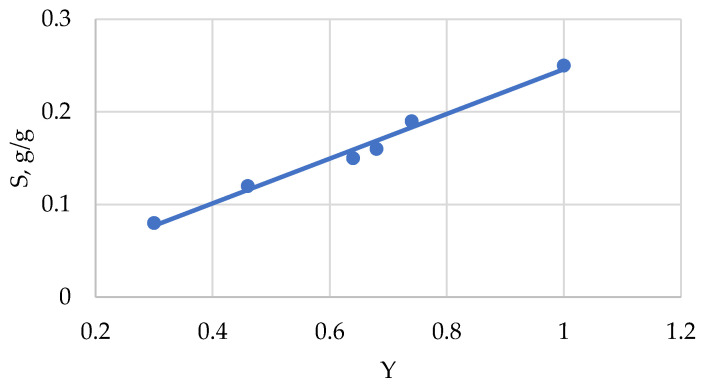
Dependence of water vapor sorption (S) at α = 0.7 on amorphicity degree (Y) of cellulose samples.

**Figure 6 polymers-13-04313-f006:**
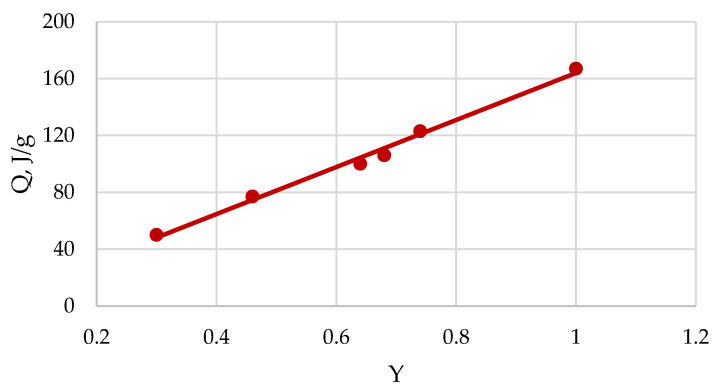
Dependence heat of wetting (Q) on amorphicity degree (Y) of cellulose samples.

**Figure 7 polymers-13-04313-f007:**
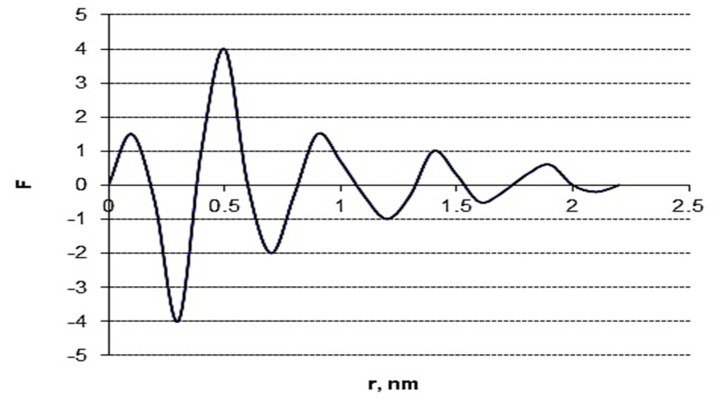
Reduced function of radial distribution of electron density.

**Figure 8 polymers-13-04313-f008:**
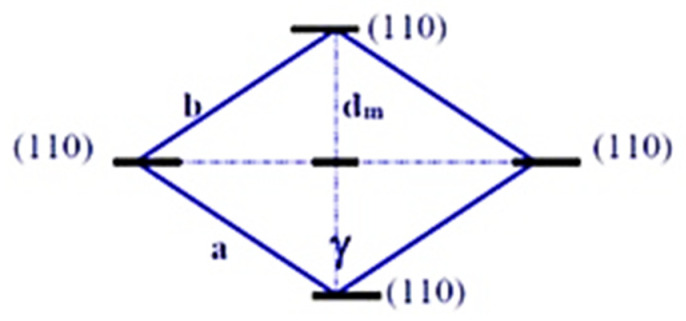
Model of mesomorphous unit cell of cluster in CAC.

**Figure 9 polymers-13-04313-f009:**
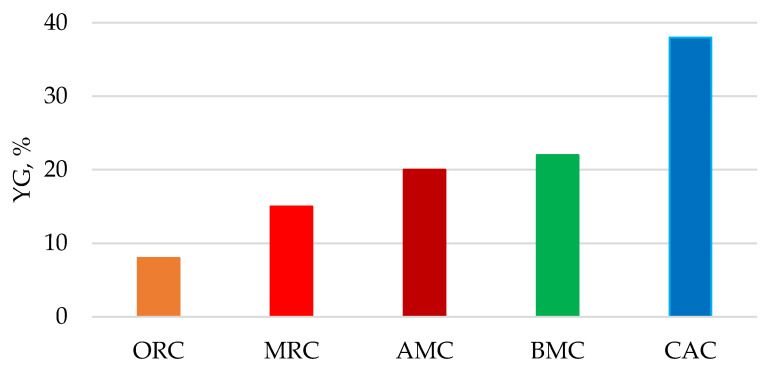
Glucose yield after acidic hydrolysis of cellulose samples.

**Figure 10 polymers-13-04313-f010:**
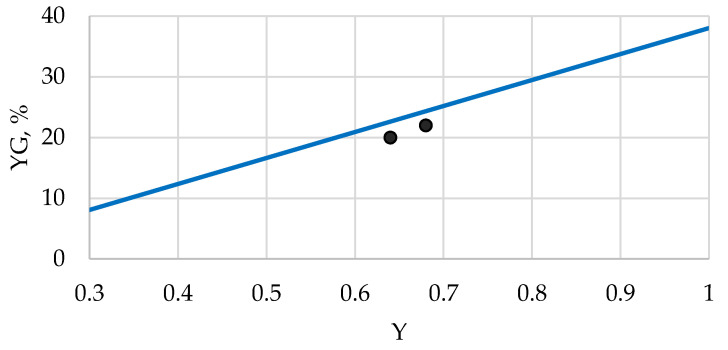
Dependence of glucose yield (YG) after acidic hydrolysis of cellulose samples on their amorphicity degree (Y) (Two dots relate to AMC and BMC samples).

**Figure 11 polymers-13-04313-f011:**
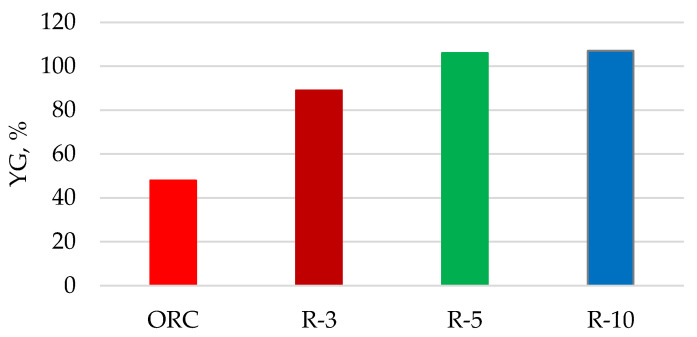
Enzymatic digestibility at C_o_ = 50 g/L of cellulose samples pretreated with cold N/U-solvent at various R-values.

**Figure 12 polymers-13-04313-f012:**
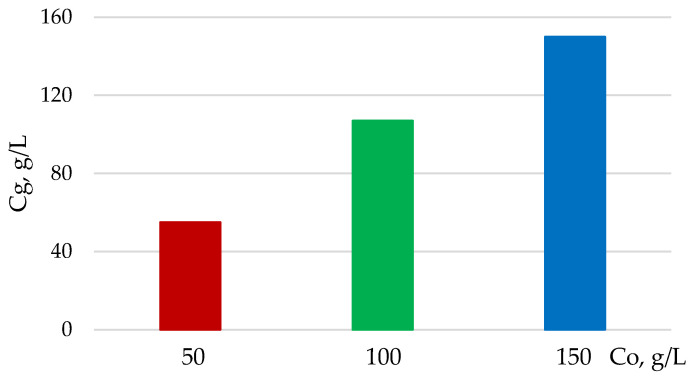
Dependence of glucose concentration (C_g_) formed after enzymatic hydrolysis on initial content of CAC substrate (C_o_).

**Figure 13 polymers-13-04313-f013:**
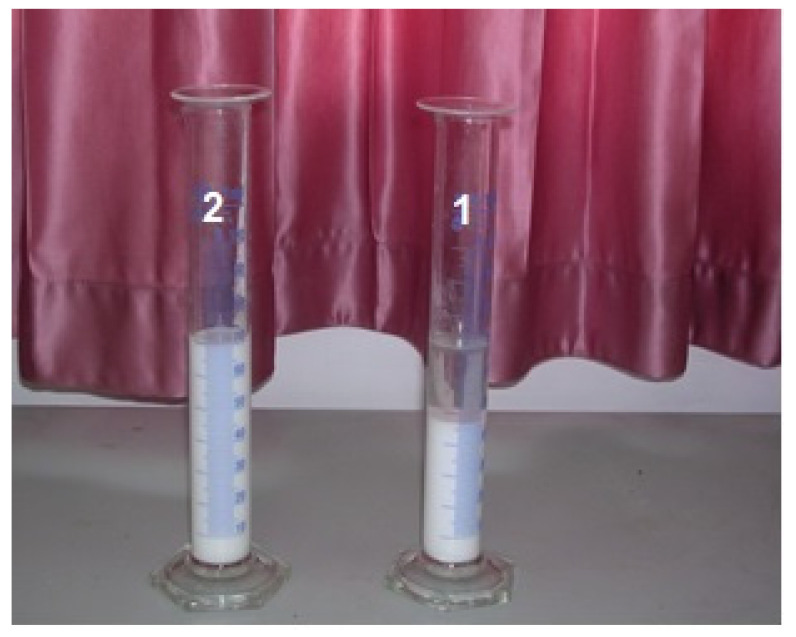
Phase separated original drug “Maalox” (**1**) and phase stable “Maalox ”containing 2% of ANC (**2**).

**Table 1 polymers-13-04313-t001:** Angular positions (2θ°_hkl_) of the main diffraction peaks and interplanar spacings (d) for different crystalline allomorphs (CA) of cellulose.

CA	Miller’s Indices	2θ°_hkl_	d, nm
CI_β_	1–10	14.5–14.7	0.6
	110	16.0–16.4	0.54
	200	22.4–22.6	0.393
	004	34.4–34.8	0.259
CII	1–10	12.0–12.2	0.73
	110	19.8–20.2	0.443
	200	21.8–22.2	0.4
	004	34.4–34.8	0.258
CIII	1–10	11.8–12.0	0.743
	110 & 200	19.8–21.2	0.423
	004	34.4–34.8	0.258
CIV	1–10 & 110	15.5–16.0	0.57
	200	22.0–22.4	0.4
	004	34.4–34.8	0.259

**Table 2 polymers-13-04313-t002:** Main features of nanoparticles of ANC.

Feature	Value
Phase state	Amorphous
Average diameter of nanoparticles, nm	cca 100
Average DP	70
Content of sulfonic groups, meq/kg	210
Content of reducing groups, meq/kg	84

**Table 3 polymers-13-04313-t003:** Structural characteristics of cellulose samples.

Amorphization Method	* AL	CrI	X	Y	D_cr_, nm
Initial sample	CI	0.66	0.7	0.3	8.3
Ball-grinding	CI	0.3	0.32	0.68	3.6
Treatment with liquid NH_3_	CIII	0.34	0.36	0.64	4.1
Mercerization	CII	0.51	0.54	0.46	5.5
Treatment with N/U solvent at					
R = 3	CII	0.23	0.26	0.74	3.4
R = 5	-	0	0	1	0
R = 10	-	0	0	1	0

* Note: AL is type of crystalline allomorph.

**Table 4 polymers-13-04313-t004:** Physico-chemical characteristics of cellulose samples.

Amorphization Methods	Y	S at α = 0.7	Q, J/g
Initial sample	0.30	0.08	50
Ball-grinding	0.68	0.16	106
Treatment with liquid NH_3_	0.64	0.15	100
Mercerization	0.46	0.12	77
Treatment with N/U solvent at			
R = 3	0.74	0.19	123
R = 5	1.0	0.24	166
R = 10	1.0	0.25	167

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
