# Peer review of "Preparation, Characterization and Application of Amorphized Cellulose—A Review"

_polymers, 2021, doi:10.3390/polym13244313_

Round 1
Reviewer 1 Report
This manuscript reports “Preparation, Characterization and Application of Amorphized Cellulose-A Review”. As in the title, this manuscript is intended to describes the methods of cellulose amorphization. However, the focal point of the manuscript is vague and the logic is not clear. Therefore, in order to benefit the readers, some statement in the manuscript need further justification. I think this article can only be re-considered after fully addressing the following major issues:
- For author information, the company location city, country, and zip code are all missing. Please update those necessary information.
- In the introduction part, I recommend the authors give more background and make a good transition between the background and the review’s unique sale point-there are numbers of reviews with the similar topic, how is this review different from others and what is the significance.
- For the section 3, I recommend the authors make it more clearly and in depth. And the section 3 should not dilute the main point of theme. This part should be the focal point of the entire manuscript.
- Sections 4 and 5 are both the analysis of characterization methods, I recommend the authors make them together.
- I recommend that the format of the figure and the table in the text behave consistently, and the centered format of the figure and the table is more convenient and more beautiful for readers.
- What is the meaning of section 7? What does the author mainly want to express?
- There are some format errors (upper and lower letters, italics) for the references, such as Ref. 5, 17, 38. Please check and revise.
- It also has some minor unit, punctuation, spelling mistakes in its article. Besides, Table 5 appears garbled.
- The authors could add the following references which would again increase the interest to general functional cellulosic material readers: Journal of Bioresources and Bioproducts, 2021, 6(1): 26-32; ACS Applied Materials & Interfaces, 2021, 13, 7617-7624; Journal of Bioresources and Bioproducts, 2021, 6(1): 75-81.
Author Response
This manuscript reports “Preparation, Characterization and Application of Amorphized Cellulose-A Review”. As in the title, this manuscript is intended to describes the methods of cellulose amorphization. However, the focal point of the manuscript is vague and the logic is not clear. Therefore, in order to benefit the readers, some statement in the manuscript need further justification. I think this article can only be re-considered after fully addressing the following major issues:
- For author information, the company location city, country, and zip code are all missing. Please update those necessary information.
Answer: These data were introduced.
- In the introduction part, I recommend the authors give more background and make a good transition between the background and the review’s unique sale point-there are numbers of reviews with the similar topic, how is this review different from others and what is the significance.
Answer: After wide literature search by Internet, I don’t find reviews in the topic of “Amorphized or Amorphous Cellulose”. There are only research articles on this topic. Some of them published in 2015-2021 I added to my review. Nevertheless, I expanded the introduction where I gave justification for the need for this review.
- For the section 3, I recommend the authors make it more clearly and in depth. And the section 3 should not dilute the main point of theme. This part should be the focal point of the entire manuscript. Answer: This section was expanded by including also methods for production of amorphous nanocellulose/
- Sections 4 and 5 are both the analysis of characterization methods, I recommend the authors make them together. Answer: It was done.
- I recommend that the format of the figure and the table in the text behave consistently, and the centered format of the figure and the table is more convenient and more beautiful for readers. Answer: It was done.
- What is the meaning of section 7? What does the author mainly want to express? Answer: In this section, various potential applications of amorphized cellulose in the form of a hydrogel have been described, which is of great practical importance. Although agricultural applications are discussed in more detail, other uses of AC gel are also mentioned, in in biology, chemistry, physical chemistry, chromatography, medicine, hygiene, pharmaceutics, food processing, water purification, and other areas.
In the revised paper, I created special section of Applications, which included various potential uses of AC and ANC. For this purpose I used content of the old section 7 and some other results.
- There are some format errors (upper and lower letters, italics) for the references, such as Ref. 5, 17, 38. Please check and revise. Answer: These ref. were corrected.
- It also has some minor unit, punctuation, spelling mistakes in its article. Besides, Table 5 appears garbled. Answer: Table 5 (now 4) was corrected. An additional check was made, and I tried to correct mistakes and typos.
-
The authors could add the following references which would again increase the interest to general functional cellulosic material readers:
Journal of Bioresources and Bioproducts, 2021, 6(1): 26-32. Title: “Plant Extract-loaded Bacterial Cellulose Composite Membrane for Potential Biomedical Applications”.
ACS Applied Materials & Interfaces, 2021, 13, 7617-7624. Title: “A Mussel-Inspired Polydopamine-Filled Cellulose Aerogel for Solar-Enabled Water Remediation”.
Journal of Bioresources and Bioproducts, 2021, 6(1): 75-81. Title: “Preparation and Properties of Cellulose Nanocomposite Fabrics with in situ Generated Silver Nanoparticles by Bioreduction Method”.
Answer: As can see from titles, the recommended references are not relevant to the review topic. However, I found other references published in 2015-2021, which I added to my review. Note: Major corrections and additions were highlighted in Yellow.
Reviewer 2 Report
Very interesting work, but in need of improvement.
Firstly, I suggest a change in the structure of the text.
Some of the citations are quite old e.g. 2001, 1993, 1982, etc.
I would suggest reviewing the work again and selecting more recent literature.
In the beginning, the author qualified it as a review, and it is difficult to say what type of work it is due to the introduction of the paragraph "Materials and methods" at the beginning.
In the section named "Materials and methods" it is difficult to determine whether these methods are authoritative. Moreover, it is not clear whether the author presents them or intends to use them.
Some of the images are pixelized, I suggest adding them in better resolution.
It is worth enriching the work with references to the application of these methods e.g. in industry and a broader description of their potential innovation.
I also suggest minor language corrections.
Author Response
Very interesting work, but in need of improvement.
Firstly, I suggest a change in the structure of the text.
Some of the citations are quite old e.g. 2001, 1993, 1982, etc. I would suggest reviewing the work again and selecting more recent literature. Answer: Additional publications for 2015-2021 I added to my review.
In the beginning, the author qualified it as a review, and it is difficult to say what type of work it is due to the introduction of the paragraph "Materials and methods" at the beginning. In the section named "Materials and methods" it is difficult to determine whether these methods are authoritative. Moreover, it is not clear whether the author presents them or intends to use them. Answer: Sorry, this was an oversight on my part. I made a correction. Now title of this section is “Characterization Methods of Original and Amorphized Celluloses”. This section describes the basic and improved methods that are currently used to characterize the structure and properties of polymers, including natural cellulose and modified cellulose with an amorphous structure.
Some of the images are pixelized, I suggest adding them in better resolution. Answer: It was done.
It is worth enriching the work with references to the application of these methods e.g. in industry and a broader description of their potential innovation. Answer: I introduced additional references. As for innovations and applications in agriculture, industry and medicine, they are described in sections 6 and 7.
I also suggest minor language corrections. An additional checking was made, and I tried to correct the language.
Note: Major corrections and additions were highlighted in Yellow.
Round 2
Reviewer 1 Report
It is Ok for me now
Reviewer 2 Report
The author has significantly improved the text according to my guidelines. My decision is to accept